# Essential Oils Derived from *Cistus* Species Activate Mitochondria by Inducing SIRT1 Expression in Human Keratinocytes, Leading to Senescence Inhibition

**DOI:** 10.3390/molecules27072053

**Published:** 2022-03-22

**Authors:** Merieme Ledrhem, Miku Nakamura, Miyu Obitsu, Kinue Hirae, Jun Kameyama, Hafida Bouamama, Chemseddoha Gadhi, Yoshinori Katakura

**Affiliations:** 1Faculty of Sciences and Techniques, Cadi Ayyad Unversity, P.O. Box 549, Bd Elkhattabi, Marrakesh 40000, Morocco; m.ledrhem@gmail.com (M.L.); bouamamahafida@gmail.com (H.B.); 2Graduate School of Bioresource and Bioenvironmental Sciences, Kyushu University, 744 Motooka, Nishi-ku, Fukuoka 819-0395, Japan; storm8114@gmail.com (M.N.); hirae.kinue.327@s.kyushu-u.ac.jp (K.H.); kameyama.jun.014@s.kyushu-u.ac.jp (J.K.); 3Graduate School of Systems Life Sciences, Kyushu University, 744 Motooka, Nishi-ku, Fukuoka 819-0395, Japan; miyu1119.ww@gmail.com; 4Faculty of Sciences Semlalia, Cadi Ayyad Unversity, P.O. Box 2390, Bd My Abdellah, Marrakesh 40000, Morocco; dgadhi@uca.ac.ma; 5Faculty of Agriculture, Kyushu University, 744 Motooka, Nishi-ku, Fukuoka 819-0395, Japan

**Keywords:** *Cistus*, essential oil, HaCaT, SIRT1, mitochondria, anti-aging

## Abstract

*Cistus* L. is a genus of dicotyledonous perennial herbaceous plants. *Cistus* species have been commonly used in folk medicine in the Mediterranean region. In the present study, the biological activities of essential oils derived from *Cistus* species *(Cistus laurifolius, C. monspeliensis, C. creticus*, and *C. salviifolius*) were evaluated. Essential oils derived from *C. laurifolius* and *C. monspeliensis* were found to augment the expression of SIRT1, an anti-aging gene, in the normal culture of HaCaT cells. Furthermore, these essential oils increased the number and size of mitochondria and augmented their activity. These effects were thought to be caused by the up- and downregulated expression of MITOL and Drp1 in HaCaT cells, respectively, in response to the essential oil treatment. In addition, these essential oils were found to attenuate ultraviolet-B-induced mitochondrial damage and cellular senescence in HaCaT cells. These findings indicate that essential oils derived from *C. laurifolius* and *C. monspeliensis* may inhibit skin aging through mitochondrial regulation via SIRT1 activation.

## 1. Introduction

*Cistus* L., commonly known as rock rose, is a genus of dicotyledonous perennial herbaceous plants that have hard leaves, grow in open areas with stony and infertile soils, and are indigenous to the Mediterranean region [1]. The family of Cistaceae consists of 8 genera and 180 species. *Cistus* species have been commonly used in folk medicine as herbal tea, extracts, and fragrances in the Mediterranean region, and specifically in Morocco [2]. Recently, phytochemical profiles and the various pharmacological activities of *Cistus* species were reviewed, where the antimicrobial activities, including antiviral, antiparasitic, antifungal, and antibacterial potentials, of essential oils, raw extracts, and isolated compounds were introduced [3]. In the present study, four *Cistus* species from the High Atlas Mountains of Morocco (*Cistus laurifolius, C. monspeliensis*, *C. creticus*, and *C. salviifolius*) which are known to be rich in phenylpropanoids and terpenes were used. Several flavonoids, including quercetin, myricetin, kaempferol, and apigenin and their aglycones, tannins, and ellagitannins, were detected in the leaves [4]. These plants contain many biologically active compounds that are expected to have a variety of functional properties, such as antioxidant activity. In the present study, the anti-aging activities of essential oils derived from the four *Cistus* species on skin keratinocytes were evaluated.

SIRT1 is a mammalian ortholog of a yeast silent information regulator 2 (Sir2). It is an NAD^+^-dependent deacetylase that mediates the effects of calorie restriction and regulates the lifespan of several organisms. SIRT1 has recently received much attention as an anti-aging gene [5]. In particular, it has been reported that SIRT1 can achieve anti-aging at the cellular level by enhancing mitochondrial biosynthesis through the activation of PGC-1α [6].

Previously, novel systems for screening food and food ingredients with anti-aging activities by targeting SIRT1 were developed. In those reports, polyphenols derived from the pomegranate were found to repair ultraviolet-B-induced DNA damage via SIRT1 activation [7,8]. Using these systems, the novel functions of essential oils derived from *Cistus* species were evaluated.

## 2. Results

### 2.1. Identification of Essential Oil That Activates SIRT1 Transcription

Among these five essential oils, essential oils derived from leaves of *C. laurifolius* before flowering, *C. salviifolius*, and *C. monspeliensis* strongly activated the SIRT1 promoter in HaCaT cells (Figure 1A), although essential oils derived from the leaves of *C. laurifolius* during flowering and *C. creticus* did not. Next, the effects of the essential oils from *C. laurifolius* before flowering and *C. monspeliensis* on the expression of endogenous SIRT1 in HaCaT cells (Figure 1B,C) were evaluated. The results showed that essential oils significantly augmented the expression of endogenous SIRT1 in HaCaT cells both at the mRNA level and at the protein level.

### 2.2. Effects of Cistus Essential Oils on the Mitochondrial Biogenesis and Differentiation

Peroxisome-proliferator-activated receptor gamma coactivator 1-α (PGC-1α) functions as a downstream factor of SIRT1 and is known to be a master regulator of mitochondrial biogenesis. As shown in Figure 2A, the essential oils increased the endogenous expression of PGC-1α in HaCaT cells. Then, the effect of essential oil on mitochondrial biogenesis was evaluated. Consistently, the essential oils increased the number and/or size of mitochondria, as evidenced by the increased copy number of mitochondrial DNA in HaCaT cells treated with these essential oils (Figure 2B). Furthermore, these essential oils augmented the expression of genes (involucrin, loricrin) relating to skin cell differentiation (Figure 2C,D), suggesting that these essential oils can enhance the function of the cell envelope and barrier.

Furthermore, in order to clarify the effects of essential oil on mitochondrial biogenesis and activity, fluorescent probes of MitoTracker Red CMXRos and MitoTracker Green FM were used. MitoTracker Red CMXRos fluorescence was dependent on mitochondrial membrane potential, whereas MitoTracker Green FM was not affected by mitochondrial membrane potential [9]. Thus, mitochondrial activity can be estimated by MitoTracker Red CMXRos fluorescence, and mitochondrial count and size can be measured using MitoTracker Green FM fluorescence by using IN Cell Analyzer 2200. An example of MitoTracker staining is shown in Figure 3A. Our results showed that these essential oils significantly activated mitochondria and increased their number and size in HaCaT cells (Figure 3B), suggesting that the essential oils regulate the morphological stability of mitochondria.

Next, the effects of these essential oils on the morphological stability of mitochondria were evaluated by investigating the expression of MITOL, a mitochondrial ubiquitin ligase, and Drp1, a mitochondrial fission factor [10]. The essential oils augmented the expression of MITOL and decreased the expression of Drp1, suggesting that they stabilize mitochondrial morphology (Figure 4A,B).

### 2.3. Effects of Cistus Essential Oils on UVB-Induced Mitochondrial Damage and Senescence

As shown in Figure 5, UVB irradiation significantly induced mitochondrial damage. UVB reduced mitochondrial activity and reduced the number and size of mitochondria in HaCaT cells. However, essential oils greatly attenuated UVB-induced mitochondrial damage in HaCaT cells.

As shown in Figure 6, UVB treatment significantly induced cellular senescence in HaCaT cells. However, essential oils significantly attenuated UVB-induced cellular senescence in HaCaT cells.

## 3. Discussion

To date, the leaf extracts and flowers of *Cistus* species have many functions in preventing various diseases and maintaining good physical condition [11,12,13,14], which may be the reasons why they are used in traditional folk medicine to improve inflammatory conditions. In the present study, essential oils derived from *Cistus* species were shown to increase mitochondrial biogenesis and activity and suppress cellular senescence induced by UVB through the activation of SIRT1. Previous studies have shown that activation of SIRT1 in skin cells repairs UV-induced DNA damage [7] and enhances the synthesis of ceramide [15,16], which is responsible for the internal skin barrier. Furthermore, it has been shown that mitochondrial activation associated with SIRT1 activation in skin cells contributes to the suppression of wrinkles [10]. Based on the above studies, the essential oil used in this study is thought to have skin improvement effects, such as skin barrier enhancement, wrinkle suppression, and repair of UVB-induced damage, through the activation of SIRT1.

Compositional analyses of these essential oils have been conducted by various researchers [17,18]. Recently, the chemical composition of the essential oil of *Cistus* species has been investigated using GC-MS and other techniques. Many sesquiterpenes and monoterpenoids have been identified, and it is thought that these components may explain the various functions including antimicrobial, anti-inflammatory, and antioxidant properties of the essential oil, but direct evidence is still scarce [1,6,19]. However, the effects of *Cistus* essential oil on skin cells, especially on skin SIRT1 and mitochondria, have not yet been studied; thus, the present study is a novel report that reveals new functions of *Cistus* essential oil on the skin. In addition, analyses to identify the components of the essential oils of *C. laurifolius* and *C. monspeliensis* involved in skin SIRT1 and mitochondrial activation are in progress, and new components that function in skin cell activation can be found in the near future.

This study revealed that the screening system used in this study is useful in the search for foods and food ingredients with anti-aging activity. Using this system, the anti-aging activities of plant extracts as well as polyphenols and lactic acid bacteria have been discovered [7,8,20]. In the present report, we used the promoter activity of an anti-aging gene as an indicator and skin cells as evaluation cells to evaluate the functionality of *Cistus* essential oil. However, if other anti-aging genes such as SIRT3 and telomerase are used as target genes and/or different cells (intestinal cells, muscle cells, brain cells) are used for screening, it will be possible to find new functionalities of *Cistus* essential oil [21].

## 4. Materials and Methods

### 4.1. Plant Materials and Essential Oil Extraction

The leaves of *C. laurifolius* specimens were collected before and during plant flowering in March 2017, from Toufliht in the High Atlas Mountains (70 km southeast of Marrakesh, Morocco). The leaves of *C. monspeliensis*, *C. creticus*, and *C. salviifolius* were collected during flowering in June 2017 from Amassine-Ourika in the High Atlas Mountains (36 km south of Marrakesh, Morocco). Voucher specimens of *C. laurifolius* (MARK-8260), *C. monspeliensis* (MARK-7724), *C. creticus* (MARK-7722), and *C. salviifolius* (MARK-9807) were deposited in the regional herbarium of the same institution.

The leaves of *C. laurifolius*, *C. monspeliensis*, *C. creticus*, and *C. salviifolius* were shade-dried and coarsely ground, and aliquots of powder (100 g) of each plant were individually used to obtain the essential oils by steam distillation for 4 h. The essential oils were recovered from the hydrolysate by liquid–liquid extraction using n-hexane. The solvent was removed under reduced pressure using a rotary evaporator at 40 °C. The essential oils were dried over anhydrous sodium sulfate and stored at 4 °C in dark-colored stoppered glasses until use. The yield of essential oil is the ratio of the weight of the extracted oil obtained after evaporation and that of the plant material used. The yields obtained from *C. laurifolius*, *C. monspeliensis*, *C. creticus*, and *C. salviifolius* were 0.127%, 0.1%, 0.078%, and 0.08%, respectively.

### 4.2. Cell Line and Treatment

The HaCaT human keratinocyte cell line (Riken Bioresource Center, Tsukuba, Japan) was cultured in Dulbecco’s modified Eagle’s medium (DMEM; Nissui, Tokyo, Japan) supplemented with 10% fetal bovine serum (FBS; Life Technologies, Gaithersburg, MD, USA) at 37 °C in a 5% CO_2_ atmosphere. Resveratrol and essential oils were dissolved in 100% DMSO at the concentrations of 10 mM and 100 μg/mL, respectively. These stock solutions were added to HaCaT cells to make a 1000-fold dilution. Therefore, in control experiments, 100% DMSO was added to HaCaT cells to make a 1000-fold dilution. Cells were treated with essential oils at the final concentration of 100 ng/mL by 1000-fold dilution of stock solution with culture medium. Resveratrol was used as a positive control to enhance SIRT1 expression in HaCaT cells.

### 4.3. SIRT1 Promoter Reporter Assay

The human SIRT1 promoter (–1593 to −1) was PCR-amplified by using human genomic DNA as a template, and its sequence was confirmed by sequencing. The amplified human SIRT1 promoter was cloned into pEGFP-C (Takara, Shiga, Japan), whose CMV promoter was removed by AseI and NheI digestion (Figure 7) [20]. The resulting plasmid (hSIRT1p-EGFP) (1 μg) was transduced into HaCaT cells (HaCaT (hSIRT1p-EGFP)) by using HilyMax (Dojindo, Kumamoto, Japan) according to the manufacturer’s protocol. Stable transformants were selected by using G418. Changes in EGFP fluorescence derived from hSIRT1-EGFP were monitored using an IN Cell Analyzer 2200 (GE Healthcare, Amersham Place, United Kingdom) [7,8].

### 4.4. Quantitative Reverse Transcription Polymerase Chain Reaction

HaCaT cells (6.0 × 10^4^ cells/mL) were seeded onto a culture dish (60 mm × 15 mm) and cultured for 24 h. The cells were further cultured for 48 h with daily addition of 10 μM resveratrol or 100 ng/mL essential oil. RNA was prepared from cells using the High Pure RNA Isolation Kit (Roche Diagnostics GmbH, Mannheim, Germany), and complementary DNA (cDNA) was prepared using the ReverTra Ace kit (Toyobo, Osaka, Japan). Quantitative reverse transcription polymerase chain reaction (qRT-PCR) was performed using the Thunderbird SYBR qPCR Mix (Toyobo) and Thermal Cycler Dice Real Time System TP-800 (Takara) [21]. Samples were analyzed in triplicate, and gene expression levels were normalized to those of β-actin. The following PCR primers were used: SIRT1, forward primer 5′-GCCTCACATGCAAGCTCTAGTGAC-3′ and reverse primer 5′-TTCGAGGATCTGTGCCAATCATAA-3′; human involucrin, forward primer 5′-TCCTCCAGTCAATACCCATCAG-3′ and reverse primer 5′-CAGCAGTCATGTGCTTTTCCT-3′; human loricrin, forward primer 5′-TCATGATGCTACCCGAGGTTTG-3′ and reverse primer 5′-CAGAACTAGATGCAGCCGGAGA-3′; human β-actin, forward primer 5′-TGGCACCCAGCACAATGAA-3′ and reverse primer 5′-CTAAGTCATAGTCCGCCTAGAAGCA-3′; human MITOL, forward primer 5′-CCCATTTACTACATTAATGCCCTTG-3′ and reverse primer 5′-CGGGCCTTCATGAGCCCTAATAC-3′; human Drp1, forward primer 5′-TCTCCAGCACATCAGATTTCAA-3′ and reverse primer 5′-TGACTCAGTTTAAGGGCCAACA-3′; human PGC-1α, forward primer 5′-GCTGACAGATGGAGACGTGA-3′ and reverse primer 5′-TAGCTGAGTGTTGGCTGGTG-3′. β-actin was used as a housekeeping gene. Samples were normalized and analyzed by the ∆∆CT method [22].

### 4.5. Western Blot

HaCaT cells (6.0 × 10^4^ cells/mL) were seeded onto a culture dish (60 mm × 15 mm) and cultured for 24 h. The cells were further cultured for 48 h with daily addition of 10 μM resveratrol or 100 ng/mL essential oil. After lysing cells using RIPA buffer, cell lysates were resolved by electrophoresis using 10% SDS-PAGE and transferred to a PVDF membrane (Amersham Hybond; GE Healthcare). The membrane was probed with anti-SIRT1 antibody (#8469 at 1:1000, Cell Signaling Technology, Danvers, MA, USA) or anti-β-actin antibody (013-24553 at 1:1000; FUJIFILM Wako Pure Chemical Corp., Osaka, Japan). Horseradish peroxidase-labeled anti-mouse IgG antibody (#7076 at 1:2000, Cell Signaling Technology) was used as a secondary antibody. The proteins were detected using an ImmunoStar LD (FUJIFILM Wako Pure Chemical Corp.) and visualized with a WSE-6100 LumnoGraph I (Atto, Tokyo, Japan).

### 4.6. DNA Isolation

HaCaT cells (6.0 × 10^4^ cells/mL) were seeded onto a culture dish (60 mm × 15 mm) and cultured for 24 h. The cells were further cultured for 72 h with daily addition of 10 μM resveratrol or 100 ng/mL essential oil. DNA was extracted from HaCaT cells using the QIAamp DNA Mini Kit (Qiagen, Valencia, CA, USA) according to the manufacturer’s instructions. DNA content was measured using a NanoDrop 2000/2000c spectrophotometer (Thermo Fisher Scientific, Waltham, MA, USA).

### 4.7. Determination of Mitochondrial DNA (mtDNA) Copy Number

qRT-PCR analyses were performed to determine the mtDNA copy number in samples using the Human Mitochondrial DNA (mtDNA) Monitoring Primer Set (Takara) in duplicates. Primers specific for SLCO2B1 and SERPINA1 were used for the determination of nuclear DNA (nDNA), and two primers (ND1, NADH dehydrogenase subunit 1, and ND5) were used for the detection of mtDNA. qRT-PCR was performed, and the mtDNA copy number was calculated according to the manufacturer’s protocol.

### 4.8. Mitochondrial Imaging

HaCaT cells (6.0 × 10^4^ cells/mL) were seeded onto a μClear fluorescence black plate (Greiner-Bio One, Tokyo, Japan) and cultured for 24 h. The cells were further cultured for 48 h with daily addition of 10 μM resveratrol or 100 ng/mL essential oil. HaCaT cells were then treated with 2.5 μM MitoTracker Red CMXRos (Molecular Probes, Eugene, OR, USA) and incubated at 37 °C for 30 min. After removing MitoTracker Red CMXRos, cells were treated with 200 nM MitoTracker Green FM (Molecular Probes) and incubated at 37 °C for 30 min. After removing MitoTracker Green FM, cells were treated with Hoechst 33342 (Dojindo) and incubated at 37 °C for 30 min. The fluorescence intensity of MitoTracker Red CMXRos, corresponding to mitochondrial activity, and the number and area of mitochondria stained with MitoTracker Green FM were analyzed using IN Cell Analyzer 2200 (GE Healthcare) and IN Cell Investigator high-content image analysis software (GE Healthcare).

### 4.9. Ultraviolet B Irradiation

Sub-confluent HaCaT cells were irradiated with ultraviolet B (UVB) at 6–14 mJ/cm^2^ using an ultraviolet crosslinker (CL-1000, UVP, Upland, CA, USA).

### 4.10. Fluorescent Senescence-Associated β-Galactosidase Assay

Irradiated or non-irradiated HaCaT cells (6.0 × 10^4^ cells/mL) were seeded onto a μClear fluorescence black plate and cultured for 24 h. The cells were further cultured for 48 h with daily addition of 10 μM resveratrol or 100 ng/mL essential oil. The fluorescent senescence-associated β-galactosidase (SA-β-Gal) assay was performed using a fluorescent substrate of β-galactosidase (C12FDG (5-dodecanoylaminofluorescein di-β-D-galactopyranoside); Setareh Biotech, Eugene, OR, USA) as previously described [20,23]. The image of each well was acquired using an IN Cell Analyzer 2200 (GE Healthcare). Hoechst 33342 staining and fluorescent SA-β-Gal staining were used to define the nuclear and whole-cell regions, and the cell number and SA-β-Gal activity were evaluated by using Developer (GE Healthcare).

### 4.11. Statistical Analysis

All experiments were performed at least three times, and the corresponding data are shown. The results are presented as mean ± standard deviation. Statistical significance was determined using a two-sided Student’s *t*-test. Statistical significance was defined as *p* < 0.05 (* *p* < 0.05; ** *p* < 0.01; *** *p* < 0.001).

## 5. Conclusions

In the present study, essential oils derived from *C. laurifolius* and *C. monspeliensis* were found to augment the expression of SIRT1 in keratinocytes, increase the number and size of mitochondria, and augment mitochondrial activity. In addition, these essential oils were found to attenuate ultraviolet-B-induced mitochondrial damage and cellular senescence in keratinocytes. These findings demonstrate that essential oils derived from *C. laurifolius* and *C. monspeliensis* may inhibit skin aging through mitochondrial regulation via SIRT1 activation.

## Figures and Tables

**Figure 1 molecules-27-02053-f001:**
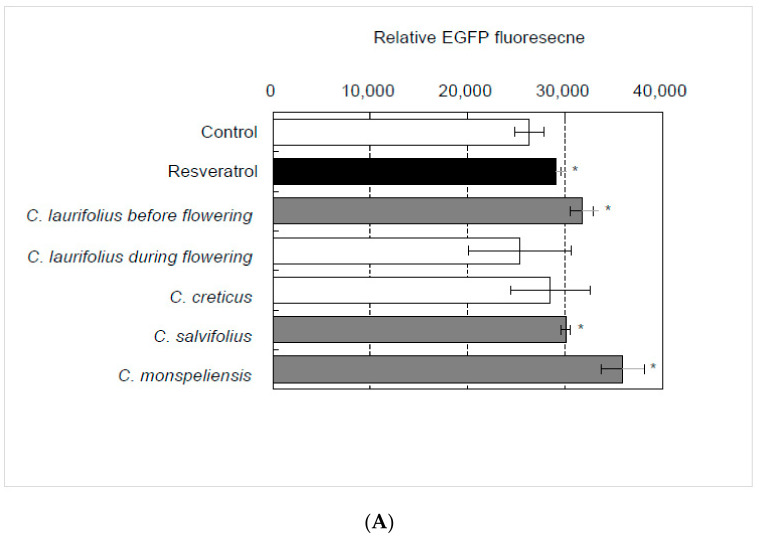
Effects of essential oils on the expression of SIRT1. (**A**) Essential oils (final concentration: 100 ng/mL) were added to the HaCaT (SIRT1p-EGFP) cells and cultured for 48 h; then, changes in EGFP fluorescence were monitored. (**B**) Effects of essential oils on the expression of endogenous SIRT1 in HaCaT cells evaluated by qRT-PCR. Statistical significance was determined by using a two-sided Student’s *t*-test. Statistical significance was evaluated by comparison with the control and defined as *p* < 0.05 (* *p* < 0.05; *** *p* < 0.001). Values are means ± SEM (*n* = 3). (**C**) Effects of essential oils on the expression of SIRT1 protein in HaCaT cells evaluated by Western blot. As a control treatment, HaCaT cells were treated with dimethyl sulfoxide (DMSO).

**Figure 2 molecules-27-02053-f002:**
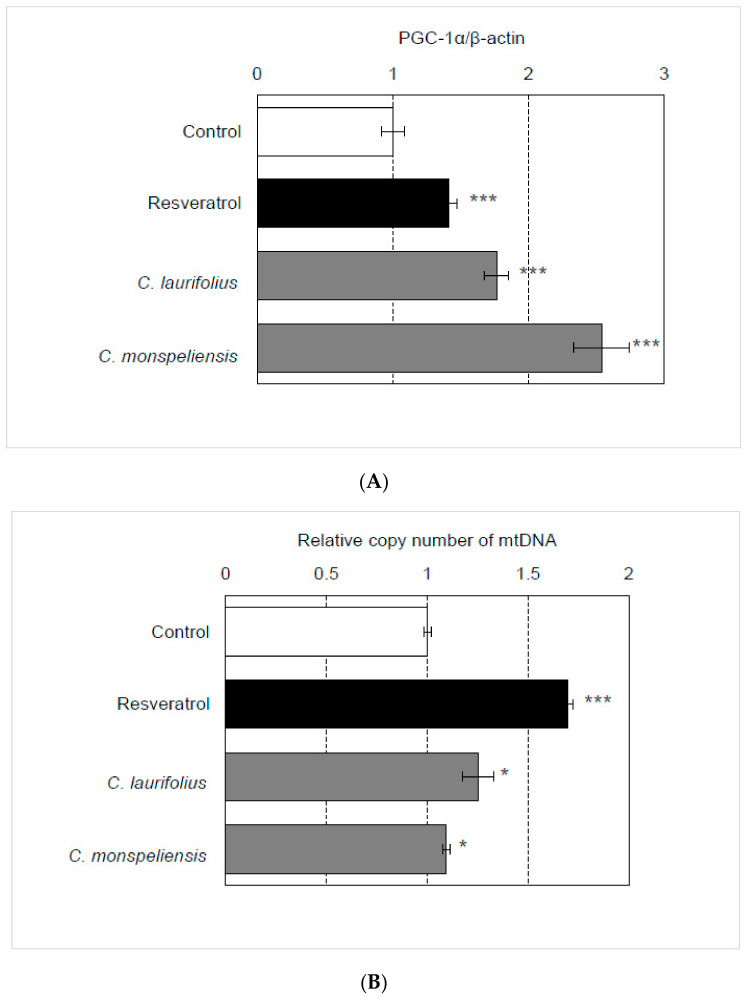
Effects of essential oils on HaCaT cells. (**A**) The effects of essential oils (final concentration: 100 ng/mL) on the expression of PGC-1α in HaCaT cells were evaluated by qRT-PCR. (**B**) The effects of essential oils (final concentration: 100 ng/mL) on the copy number of mitochondria DNA in HaCaT cells were evaluated by qPCR. (**C**,**D**) The effects of essential oils (final concentration: 100 ng/mL) on the expression of involucrin and loricrin in HaCaT cells were evaluated by qRT-PCR. Statistical significance was determined by using a two-sided Student’s *t*-test. Statistical significance was evaluated by comparison with the control and defined as *p* < 0.05 (* *p* < 0.05; *** *p* < 0.001). Values are means ± SEM (*n* = 3). As a control treatment, HaCaT cells were treated with DMSO.

**Figure 3 molecules-27-02053-f003:**
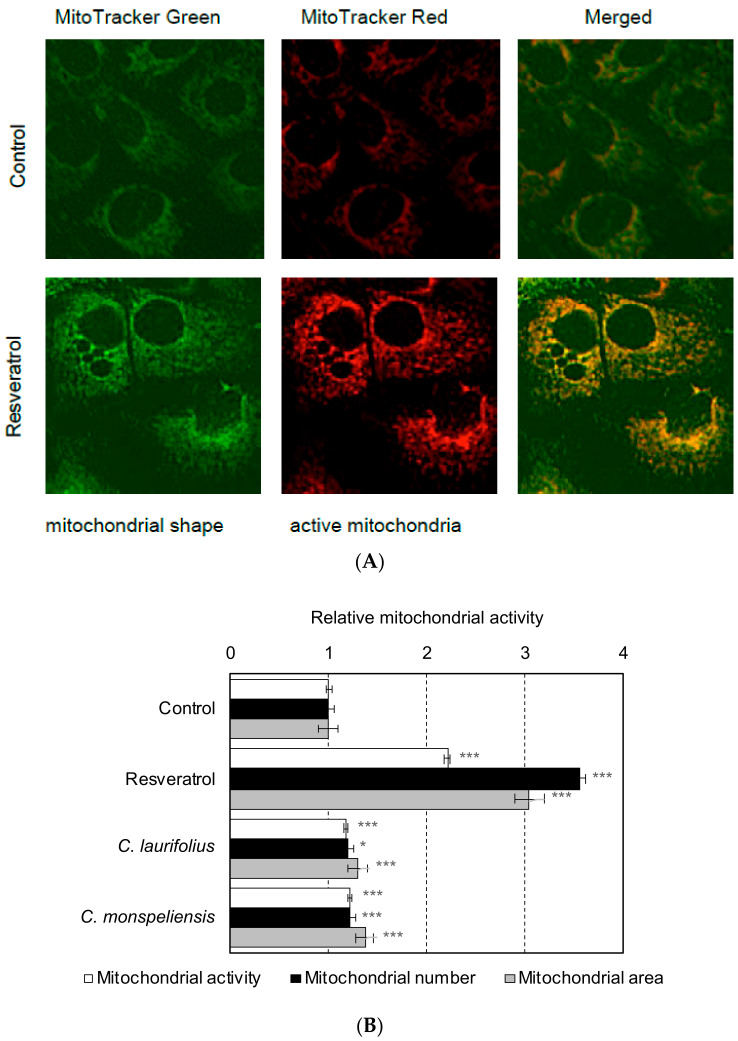
Effects of essential oils on mitochondria evaluated by fluorescence imaging. (**A**) Non-treated and resveratrol-treated HaCaT cells were stained with MitoTracker Red CMXRos and MitoTracker Green FM. (**B**) The effects of essential oils (final concentration: 100 ng/mL) on mitochondria were evaluated by using fluorescence probes of MitoTracker Red CMXRos and MitoTracker Green FM. The fluorescence intensity of MitoTracker Red CMXRos, corresponding to mitochondrial activity, and the number and area of mitochondria stained with MitoTracker Green FM were analyzed using IN Cell Analyzer 2200 (GE Healthcare) and IN Cell Investigator high-content image analysis software (GE Healthcare). Statistical significance was determined by using a two-sided Student’s *t*-test. Statistical significance was evaluated by comparison with the control of each experiment and defined as *p* < 0.05 (* *p* < 0.05; *** *p* < 0.001). Values are means ± SEM (*n* = 3). As a control treatment, HaCaT cells were treated with DMSO.

**Figure 4 molecules-27-02053-f004:**
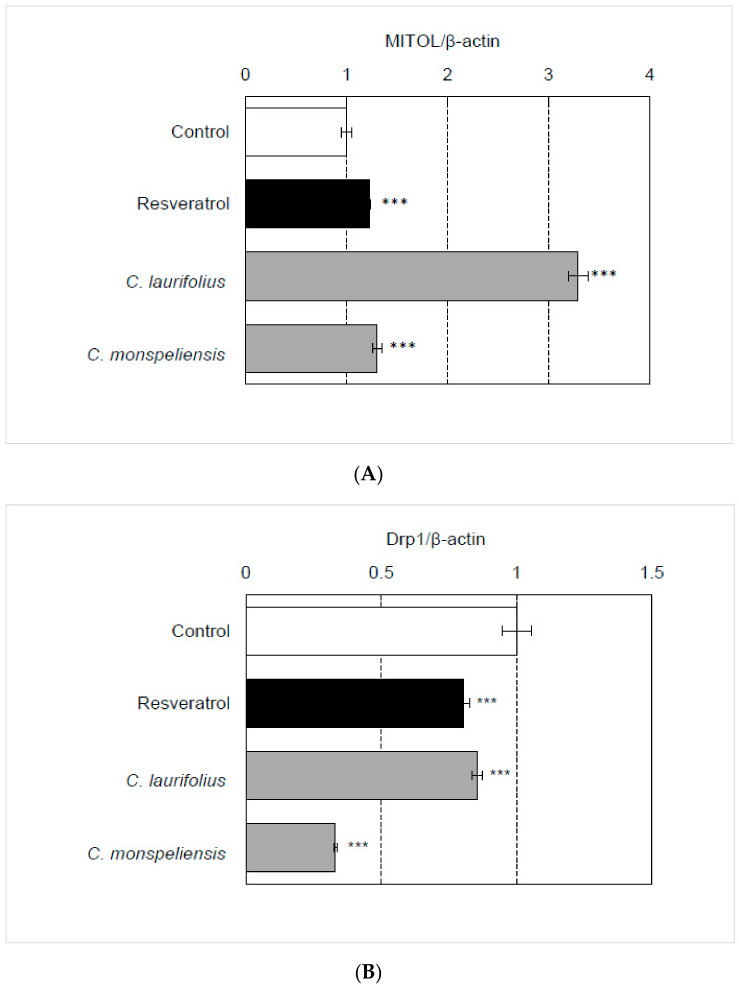
Effects of essential oils on the gene expression involved in the regulation of mitochondrial morphology. The effects of essential oils (final concentration: 100 ng/mL) on the expression of MITOL (**A**) and Drp1 (**B**) in HaCaT cells were evaluated by qRT-PCR. Statistical significance was determined by using a two-sided Student’s *t*-test. Statistical significance was evaluated by comparison with the control and defined as *p* < 0.05 (*** *p* < 0.001). Values are means ± SEM (*n* = 3). As a control treatment, HaCaT cells were treated with DMSO.

**Figure 5 molecules-27-02053-f005:**
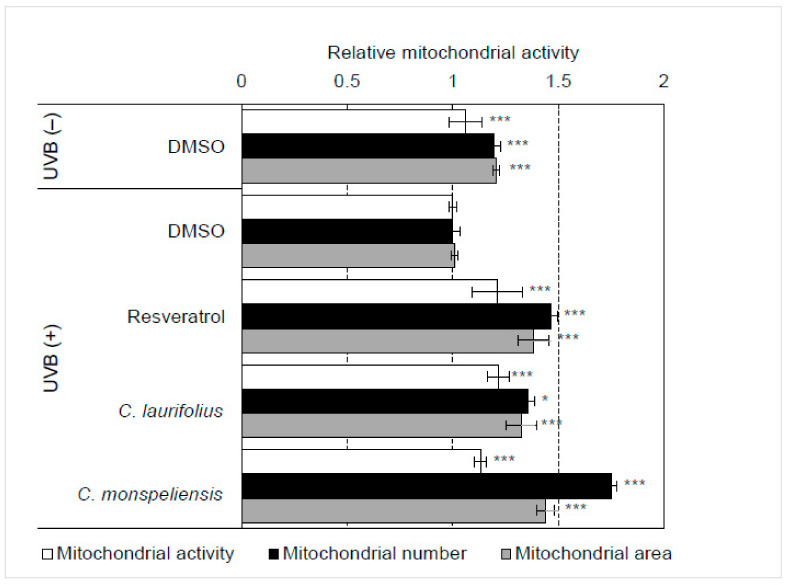
Effects of essential oils on UVB-induced mitochondrial damage. HaCaT cells were irradiated with 10 mJ/cm^2^ and then treated with essential oils (final concentration: 100 ng/mL) for 48 h. The effects of essential oils on the UVB-induced mitochondrial damage were evaluated by fluorescence imaging using MitoTracker Red CMXRos and MitoTracker Green FM. Mitochondrial activity and the number and area of mitochondria were analyzed using IN Cell Analyzer 2200 (GE Healthcare) and IN Cell Investigator high-content image analysis software (GE Healthcare) as mentioned above. Statistical significance was determined by using a two-sided Student’s *t*-test. Statistical significance was evaluated by comparison with the control irradiated with UVB and defined as *p* < 0.05 (* *p* < 0.05; *** *p* < 0.001). Values are means ± SEM (*n* = 3).

**Figure 6 molecules-27-02053-f006:**
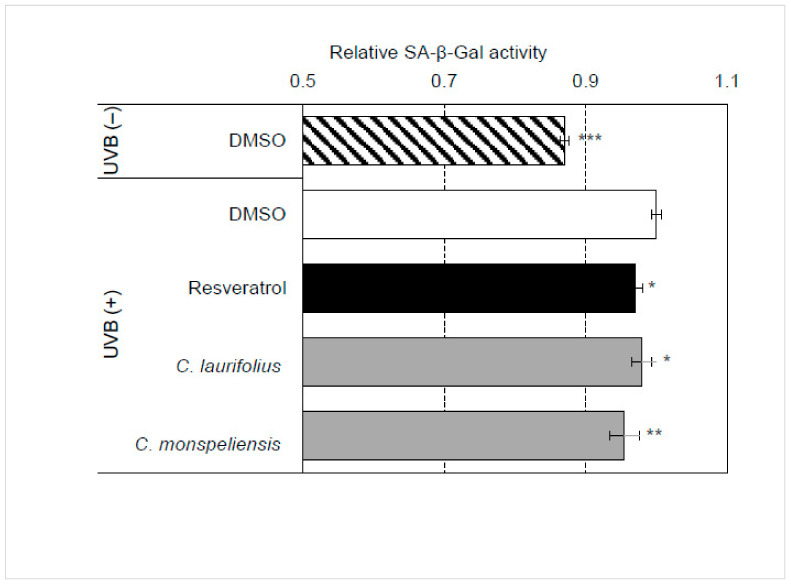
Effects of essential oils on UVB-induced cellular senescence. HaCaT cells were irradiated with 10 mJ/cm^2^ and then treated with essential oils (final concentration: 100 ng/mL) for 48 h. The effects of essential oils on the UVB-induced cellular senescence were evaluated by using a fluorescent substrate of β-galactosidase, C12FDG. Statistical significance was determined by using a two-sided Student’s *t*-test. Statistical significance was evaluated by comparison with the control irradiated with UVB and defined as *p* < 0.05 (* *p* < 0.05; ** *p* < 0.01; *** *p* < 0.001). Values are means ± SEM (*n* = 3).

**Figure 7 molecules-27-02053-f007:**
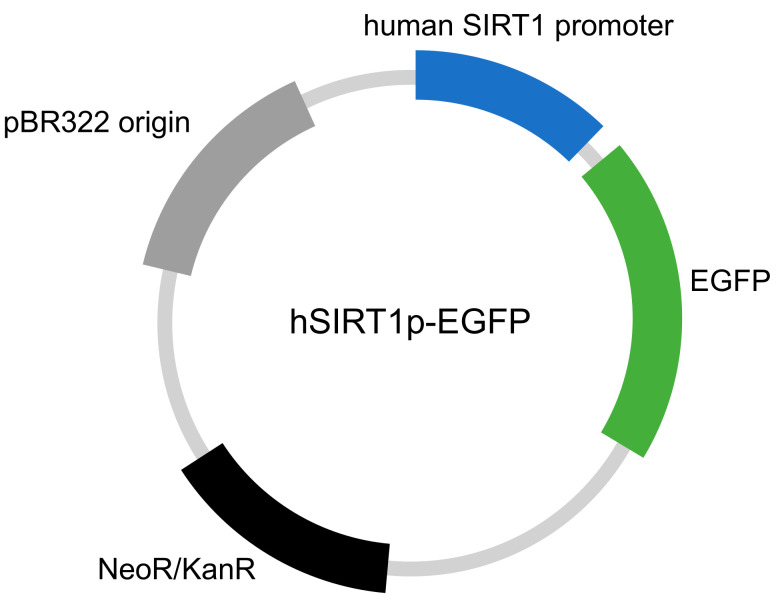
SIRT1 promoter reporter vector.

## Data Availability

The data presented in this study are openly available in Kyushu University Institutional Repository (QIR).

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
