# Peer review of "Essential Oils Derived from Cistus Species Activate Mitochondria by Inducing SIRT1 Expression in Human Keratinocytes, Leading to Senescence Inhibition"

_molecules, 2022, doi:10.3390/molecules27072053_

Round 1

Reviewer 1 Report

Please see the attachment for details.

Author Response

Reviewer #1

Comment #1:

The title does not mean a clear conclusion, for example, it says, Essential oils derived from Cistus species activate mitochondria by inducing SIRT1 expression in human keratinocytes; but it does not say why the mitochondrial activation is important or what is happening after this activation, good or bad etc.

Response #1:

In response to the comment, we revised the title.

Comment #2:

Figure 1C. There is nothing visible. In figure legends, As a control treatment, HaCaT cells were treated with dimethyl sulfoxide (DMSO). Why DMSO was used? 100% DMSO? Were the oil samples dissolved into DMSO? If so, was the DMSO content kept same for all samples? Please check everywhere.

Response #2:

We replaced the figure.

Resveratrol and essential oils were dissolved in 100% DMSO at the concentration of 10 mM and 100 μg/mL, respectively. These stock solutions were added to HaCaT cells to make a 1000-fold dilution. Therefore, in control experiments, 100% DMSO was added to the cells to make a 1000-fold dilution.

The Materials and Methods section has been revised accordingly.

Comment #3:

Line 98-99, Furthermore, these essential oils augmented the expression of genes (Involucrin, Loricrin) relating to skin cell differentiation (Figure 2C and D). Please mention what differentiation, from which cell types to what cell types etc.

Response #3:

In response to the comment, we revised the sentence.

Comment #4:

In all graphs, *, **, *** were used to show significance, but don’t mention compared to which conditions.

Response #4:

In response to the comment, we revised the figure legends.

Comment #5:

Figure 3B, please provide all separate colored and merged images mentioning each condition and color representing what, show the differences using arrows or something like that, provide magnified images of just 1 or 2 cells for better illustration.

Response #5:

In response to the comment, we revised the Figure 3A.

Comment #6:

Conclusion is very bold based on the experimental data. Please write according to the data.

Response #6:

In response to the comment, we revised the discussion.

Comment #7:

Figure 4A, please remove the 1 from the control if not required.

Response #7:

We revised the Fig. 4A.

Comment #8:

Figure 5, all data showing significant sign, but not mentioning compared to which condition.

Response #8:

In response to the comment, we revised the figure legend.

Comment #9:

English corrections by professional English editors from biological expertise required. For example, line 191-194, the sentence could be written in different way to clarify what authors were trying to say.

Response #9:

Thank you for your comment. In response to the comment, we revised the sentence.

Comment #10:

Line 251-253, Cells were treated with essential oils at the final concentration of 100 ng/mL by 1000-fold dilution of stock solution with culture medium. Then why DMSO was used as a control?

Response #10:

Thank you for your comment. We revised the Materials and Methods for clarity.

Comment #11:

In 4. Materials and Methods, the use of 10 μM resveratrol or 100 ng/mL essential oil were mentioned several times without mentioning the reason. Please mention why they were used, for example to induce differentiation or express certain gene expression etc.

Response #11:

In response to the comment, we revised the Materials and Methods.

Comment #12:

Please mention the concentrations or dilution factors for all antibodies used.

Response #12:

In response to the comment, we revised the Materials and Methods.

Comment #13:

Ultraviolet B irradiation. Sub-confluent HaCaT cells were irradiated with ultraviolet B (UVB) at 6 – 14 mJ/cm2 using an ultraviolet crosslinker (CL-1000, UVP, Upland, CA, USA). Please mention for how long UVB were used to treat the cells.

Response #13:

Thank you for your comment.

The ultraviolet crosslinker (CL-1000) automatically sets the irradiation time and irradiates according to the intensity of UVB.

Reviewer 2 Report

The manuscript was considerably improved, the figures were corrected as suggested, and the data they added seems complete and understandable to me, they also added important statistical data in the figures.

Author Response

Thank you for your evaluation.

Reviewer 3 Report

Authors tried to modify the manuscript according to the reviewers' comments. However, there are still questions about the experiments. 

Besides of all the other parts (introduction, discussion, materials and methods sections), results parts need to be reconsidered for supporting the idea of the manuscript. 

a. As a major concern, RNA and protein level need to be compared to support the idea. Once keratinocytes are differentiated by any stimulation, cells highly express differentiated markers including involucrin, loricrin. However, the data of qRT-PCR doesn't reflect differentiation markers expression. Moreover, RNA level expression of certain gene is not correlated with their protein expression level. Therefore, if the oil treatment was involved in cell differentiation, these markers' expression level should be increased way more than presented in the figure. 

b. Figure 1B: Mathematically, 0.2 vs 0.8 and 2 vs 8 comparison is same as 4 folds difference. However, biologically comparing 0.2 vs 0.6 value is not the case. Moreover, in qPCR data, to show increasing expression level of certain gene, the value should be shown above 2 folds which data will be reliable. Otherwise, below the value of 1 comparison is hard to accept. 

c. Figure 1C: the data is not shown in my case. Just black (SIRT1) and grey (beta-actin) background. Please, check the uploaded data again. 

Author Response

Reviewer #3

Comment a:

As a major concern, RNA and protein level need to be compared to support the idea. Once keratinocytes are differentiated by any stimulation, cells highly express differentiated markers including involucrin, loricrin. However, the data of qRT-PCR doesn't reflect differentiation markers expression. Moreover, RNA level expression of certain gene is not correlated with their protein expression level. Therefore, if the oil treatment was involved in cell differentiation, these markers' expression level should be increased way more than presented in the figure.

Response a:

As the reviewer pointed out, I believe it is true that changes at the RNA level do not reflect changes at the protein level.

However, in this study, we examined the effect of essential oil not only on the expression of RNA level, but also on mitochondria and cellular senescence level of each cell using an imaging cytometer, and found that the effect of essential oil was more pronounced. We believe that we have been able to verify this with high sensitivity and accuracy.

Thank you for your understanding.

Comment b:

Figure 1B: Mathematically, 0.2 vs 0.8 and 2 vs 8 comparison is same as 4 folds difference. However, biologically comparing 0.2 vs 0.6 value is not the case. Moreover, in qPCR data, to show increasing expression level of certain gene, the value should be shown above 2 folds which data will be reliable. Otherwise, below the value of 1 comparison is hard to accept.

Response b:

Thank you for your comment. In response to the comment, we replaced the Fig. 1B with new one.

Comment c:

Figure 1C: the data is not shown in my case. Just black (SIRT1) and grey (beta-actin) background. Please, check the uploaded data again.

Response c:

Thank you for your comment. In response to the comment, we replaced the Fig. 1C with new one.

Reviewer 4 Report

Minor remarks

  • All minor remarks are highlighted in the manuscript.

Major remarks

  • In the manuscript the authors should avoid using the first person plural.

Author Response

Reviewer #4

Thank you for your valuable comments.

In response to the comments, we revised the manuscript.

Round 2

Reviewer 1 Report

Please provide a representative image for Figure 1 (SIRT1/β-actin 1 1.15 1.26 1.31).

Figure 5 shows lower mitochondrial activity in UVB (+) compared to UVB (-). Treatment with essential oils increased the mitochondrial activity much more than the UVB (-) group. Is that statistically significant compared to the UCB (-) group? Please mention this in the figure.

Author Response

Reviewer #1

Comment #1:

Please provide a representative image for Figure 1 (SIRT1/β-actin 1 1.15 1.26 1.31).

Response #1:

The values for SIRT1/β-actin were calculated by quantifying the respective band intensities with LuminoGraph.

In response to the comment, the figure legend has been revised.

Comment #2:

Figure 5 shows lower mitochondrial activity in UVB (+) compared to UVB (-). Treatment with essential oils increased the mitochondrial activity much more than the UVB (-) group. Is that statistically significant compared to the UCB (-) group? Please mention this in the figure.

Response #2:

In response to the comment, we revised the figure and figure legend.

Reviewer 3 Report

.

Author Response

Thank you for your valuable comments.

This manuscript is a resubmission of an earlier submission. The following is a list of the peer review reports and author responses from that submission.

Round 1

Reviewer 1 Report

Minor remarks

The scientific manuscript should be written using the third-person singular. Please, avoid the use of first-person plural (we do that, etc.).

All other minor remarks are given in the manuscript.

Major remarks

Provide a better literature review in the Introduction section.

Since the topic of the manuscript was to investigate the biological activity of essential oils it is desirable to present the results of their chemical composition defined by the GC-MS. In this way, the quality of the manuscript will be improved. I recommend providing these results if possible.

Reviewer 2 Report

Please see the attachment for details.

Reviewer 3 Report

It is interesting data focusing on anti-aging effect of plant oil on HacaT cells. However, there are several concerns about the manuscript.

a. How did you dissolve the oil component in the cell media. As media is made based on water, oil treatment is not easy. If you dissolve the material 100% in the media, please describe the way in the method section. 

b. As authors focused on anti-aging effect of their oil on HacaT cells, did you check any differentiation differences after treatment? If not, please check differentiation of HacaT cells after treatment (morphological changes, differentiation markers expression, etc).

c. As you check on qPCR for SIRT1 expression by promoter activity, did you recognize any expression difference by western blotting. Since RNA expression does not represent protein expression all the time, please, add protein expression level in the figure 1. 

d. Authors investigated mitochondrial potentials. Please, add picture of morphological changes. 

e. It seems that resveratrol was used as a positive control, but the comparing materials are oils. Therefore, it would be nice to use any oil components to compare authors' hypothesis. Clove oil or other oil materials are recommended. 

Reviewer 4 Report

I reviewed the manuscript entitled: Essential oils derived from Cistus species activate mitochondria 2 by inducing SIRT1 expression".

General comments:

This could be an interesting research work related to the “Essential oils derived from Cistus species activate mitochondria 2 by inducing SIRT1 expression.

Unfortunately, this manuscript lacks many important data for the correct interpretation of the same.

The introduction section should be improved by placing more emphasis on the subject of the SIRT1 gene and the mitochondria.

The results describe the method but do not sufficiently describe the results obtained, so it should be improved.

The methodology section does not describe the test with essential oils, it does not specify how long it was left with the cell, when it was withdrawn from the culture and when the measurement was taken. Please add this in the method section.

The method do not describe the control used in the assays, please indicate.

The scientific names of the figures must be in italics.

It is not described how the expression levels of the gene were calculated. Please describe and mention the equation utilized if apply.

The discussion must be improved, the results are not discussed, but things that were not done in this work are discussed, please discuss the results obtained from the work presented.